# Socio-ecological network structures from process graphs

Angelyn Lao[1]*, Heriberto Cabezas[2,3], Ákos Orosz[4], Ferenc Friedler[3,5], Raymond Tan[6]

**1** Mathematics and Statistics Department, De La Salle University, Manila, Philippines, **2** University of Miskolc, Research Institute of Applied Earth Science, Miskolc, Hungary, **3** Institute for Process Systems Engineering and Sustainability, Pázmány Péter Catholic University, Budapest, Hungary, **4** Department of Computer Science and Systems Technology, University of Pannonia, Veszprém, Hungary, **5** Széchenyi István University, Debrecen, Hungary, **6** Chemical Engineering Department, De La Salle University, Manila, Philippines

* angelyn.lao@dlsu.edu.ph

**Data Availability Statement:** All relevant data are within the manuscript and its Supporting Information files.

**Funding:** This work was supported by the Pázmány Péter Catholic University Central Funds Program to HC, which allowed for the conduct of discussions

## Abstract

We propose a process graph (P-graph) approach to develop ecosystem networks from knowledge of the properties of the component species. Originally developed as a process engineering tool for designing industrial plants, the P-graph framework has key advantages over conventional ecological network analysis techniques based on input-output models. A P-graph is a bipartite graph consisting of two types of nodes, which we propose to represent components of an ecosystem. Compartments within ecosystems (e.g., organism species) are represented by one class of nodes, while the roles or functions that they play relative to other compartments are represented by a second class of nodes. This bipartite graph representation enables a powerful, unambiguous representation of relationships among ecosystem compartments, which can come in tangible (e.g., mass flow in predation) or intangible form (e.g., symbiosis). For example, within a P-graph, the distinct roles of bees as pollinators for some plants and as prey for some animals can be explicitly represented, which would not otherwise be possible using conventional ecological network analysis. After a discussion of the mapping of ecosystems into P-graph, we also discuss how this framework can be used to guide understanding of complex networks that exist in nature. Two component algorithms of P-graph, namely maximal structure generation (MSG) and solution structure generation (SSG), are shown to be particularly useful for ecological network analysis. These algorithms enable candidate ecosystem networks to be deduced based on current scientific knowledge on the individual ecosystem components. This method can be used to determine the (a) effects of loss of specific ecosystem compartments due to extinction, (b) potential efficacy of ecosystem reconstruction efforts, and (c) maximum sustainable exploitation of human ecosystem services by humans. We illustrate the use of P-graph for the analysis of ecosystem compartment loss using a small-scale stylized case study, and further propose a new criticality index that can be easily derived from SSG results.

among the authors of this manuscript. The funder had no role in study design, data collection and analysis, decision to publish, or preparation of the manuscript.

**Competing interests:** The authors have declared that no competing interests exist.

## Introduction

Mathematical models have proven to be valuable and useful tools for the analysis of ecological networks and their emergent properties. Early examples include input-output models similar to those used to describe economic structures [1]. Metrics to describe the structure of ecological networks naturally flowed from the use of such quantitative tools (e.g., [2]). These tools provide a lens for the analysis of complex interactions that arise from interactions among ecosystem components. In many cases, specialists only fully understand local interactions of ecosystem components, and thus need modelling techniques such as ecological network analysis to deduce high-level interactions that occur through direct and indirect linkages.

Searching the Scopus database using "ecological network analysis" as a search term yields 476 published documents, over half of which were published from 2016 to the present. Despite the broad array of techniques already used in ecological network analysis, according to Poisot et al. [3], "ecology will probably continue to benefit from those tools, metrics and models developed in other fields." Thus, in this paper, we discuss a potential new tool for ecological network analysis.

The use of network-based techniques for the analysis of social-ecological interdependencies remains a challenge [4]. Such models are extended from ecological network analysis methods through linkage with a network model of a human community at an appropriate scale. Network techniques are useful for understanding emergent behavior that arises from complex interactions among system components [5]. Such system-level behavior is often not immediately evident from the local properties of individual components, and failure to account for them can often lead to unexpected results [3].

On the other hand, judicious use of ecological network analysis and extensions that link them to man-made systems can provide useful insights for sustainable use or resources and ecosystem services [6–9]. These services refer to the conditions and processes by which ecosystems sustain human life [10], which can be achieved for example through the provision of shelter, nectar, alternative prey/host and/or pollen for natural enemies which can be deployed by humans [11, 12]. The insights drawn can be used to guide decisions on ecosystem conservation or restoration measures. By incorporating ecological-economic interactions, the models can also be used to estimate the limits of exploitation of ecosystem products and services.

Two challenges are apparent in the current literature on social-ecological models. First, existing techniques must assume that a single type of interaction predominates in the system. For example, trophic linkages in food webs are the most commonly represented type of relationship in ecological network models. In order to better understand the behavior of real ecosystems, the capability to represent the existence of multiple simultaneous interdependencies is needed [13]. The current approach relies on multiplex network modelling approaches—i.e., the use of multiple linked network models, each representing one type of interdependence [4, 14].

The second challenge is network assembly. Typically, model developers use best available scientific knowledge of local interactions of system components, coupled with heuristic network assembly rules, to deduce the structure of the network [15]. While this technique appears to work reasonably well, a mathematically rigorous approach to network assembly can improve ecological modelling by eliminating the potential for human error and biases that always exists when a heuristic is used.

The structure of even relatively small ecosystems can be rather complex. Any interactions that could not be observed will, therefore, often not be included. Subtle or unexpected interactions can be discovered by biologists which change the way a species relates to the rest of the ecosystem; for example, it has been reported recently that tracks of large herbivores act as

micro-habitats for small animals [16]. Such direct interactions can also lead to indirect "ripple effects" that are difficult to deduce except through the lens of an appropriate modelling technique. One should also note that frequently more is known about the individual species making up the ecosystem rather than the structure of the ecosystem itself. For example, zoologists are usually able to identify staple food sources of animal species with good accuracy. The reason is that it is far easier to study one species at a time than many species simultaneously. The challenge is to assemble such pockets of scientific knowledge into a coherent global picture of the ecosystem. Most ecological network analysis work then gives fragmented or one-dimensional representations of real ecosystems. Whereas in reality, ecosystem components play multiple roles relative to each other; thus, there is a need for a framework that allows concurrent modelling of these multiple roles and their resulting complex interactions [17].

While there has been a concerted effort by a small number of researchers to be able to track and measure the complexity of ecological systems, the common approaches in systems ecology still limit their focus to biomass (or energy, nutrients, etc.) fluxes between species (nodes), recycling of material, decomposition, or production. The species are treated as compartments that are interconnected by transaction of the energy–matter substance flowing between them. More subtle interactions such as, for example, the provision of shelter are difficult to include within this framework.

We propose the use of a class of models known as process graphs to deal with these difficulties. The process graph (or P-graph) framework was originally developed as a graph theoretic technique for handling combinatorial challenges in industrial plant design [18, 19]. In such engineering problems, the typical task is to design an optimal plant subject to economic constraints to produce a set of specified products from available raw materials, using a predefined set of candidate conversion processes. These process units must be assembled into a feasible network of conversion processes that can be then translated into a viable industrial plant. This problem is known as process network synthesis (PNS). In this work, the process of deducing an ecosystem network structure from the current scientific knowledge on its different components is treated as a PNS-like problem.

In PNS, a P-graph is a bipartite graph with two sets of nodes used to represent processes and streams. Arcs are used to denote the relationship of the inputs and outputs of material or energy streams into each of the candidate processes. The key feature of the P-graph framework is the availability of algorithms [19] developed rigorously based on a set of axioms that apply to all PNS problems [18]. The MSG and SSG algorithms are also described briefly in S3 Appendix. For any given plant design task, these algorithms can be used to automatically generate a maximal structure that represents the union of all possible process networks, and also to identify every structurally feasible process network. They eliminate the risk of human error or bias in specifying an incomplete process network during engineering design.

The P-graph framework has been applied to a wide range of PNS problems, as well as to other engineering problems of analogous form [20]. The basic mathematical machinery is independent of the nature of the nodes and links between the nodes, and it is, therefore, applicable in principle to any network. However, there is no record of its use for ecological systems in the scientific literature.

P-graph can be used to represent a class of socio-ecological systems where human society acts an external consumer of different ecosystem services. Various agents and compartments can be represented as process nodes (O-type nodes), as in conventional network approaches. These nodes can be linked to each other via different types of relationships, such as predation among species or consumption of resources and ecosystem services by humans. The different roles or functions played by the process nodes in the network can then be represented as intermediate nodes (M-type nodes). In P-graph the arcs never link process nodes directly to each

other; instead, the arcs are linked to intermediate nodes to unambiguously specify the exact nature of the relationship between any two components of the social-ecological network. This feature allows different types of interactions to be represented in a single graph, which is a departure from the conventional multiplex graph approach [4, 14].

In [21], P-graph and Petri nets are compared and combined. They showed the correspondence between P-graph and Petri nets, and applied the algorithms (MSG and SSG) to P-graphs converted from Petri nets. In the hybrid P-graph and Petri net methodology, the study illustrates the potential of P-graph to be used in conjunction with other mainstream modelling techniques. Although these two approaches have been developed for different purposes, they both use the same structural representation—bipartite directed graphs. The vertices of P-graph are the elements of $V = M \cup O$, where materials ($M$) and operating units ($O$) are disjoint sets. The nodes of a Petri net are the elements of $N = P \cup T$, where places ($P$) and transitions ($T$) are disjoint sets. Operating units and materials of P-graph correspond to transitions and places that are defined in a Petri net. The reason for representing them using a bipartite graph is, however, rather different. Despite the similarities between P-graph and Petri nets, the target of application of these two approaches are different. P-graphs optimize the desired targets of the process systems using the available resources. P-graphs also generate the optimal structure and optimal parameter values of process systems. P-graphs are applied to steady state processing systems. In contrast, Petri nets are applied primarily for modeling concurrent and asynchronous distributed dynamical systems.

Multilayer networks emerge as a framework to study complex ecological systems that account for multiple interaction types and multiple layers of complexity, such as the heterogeneous nature of the systems. In an ecological context, different network layers commonly represent different types of interactions, different communities of species, different points in time, and so on [22, 23]. A multilayer graph $G_M = (V_M, E_M)$ contains set of nodes $V_M$ that is represented by $V \times L_1 \times \cdots \times L_d$ with layers $L_i$ where $i = 1, \cdots, d$; and a set of edges $E_M$ that is defined by $V_M \times V_M$. Unlike P-graph that is mainly applied to steady state processing systems and optimize desired targets of the process systems, multilayer networks approach is mainly applied for modeling complex dynamical systems. Even though multilayer networks have been widely applied in ecology, there is a huge potential in developing the applications of P-graphs in ecology for different target approaches. To our knowledge, there is no correspondence study done for P-graphs and multilayer networks.

In addition to enabling a more elegant system representation, the P-graph framework can also be used to generate a complete socio-ecological network based only on knowledge of local interactions of each network component. The maximal structure generation (MSG) and solution structure generation (SSG) algorithms can replace the conventional heuristic approach which still runs the risk of yielding an incorrect, incomplete or biased network [15]. The capability to identify all structurally feasible networks for a given set of compartments and agents has many potential applications in sustainable ecosystem management. For example, it becomes possible to identify which structures can be functional, and which ones will collapse. Keystone species can be identified based on the frequency of occurrence in different solution structures. In direct analogy with the industrial plant design problem for which P-graph was originally developed, it is possible in principle to determine the minimum ecological cost of providing a given set of resource outputs and ecosystem services for a community of humans. This type of problem can be applied concretely to cases such as the management of fisheries, forests, and the like. The list of other possible problems extends well beyond these examples.

We show the rich potential of the P-graph to deal with the analysis of a simple socio-ecological network model. This technique has already proven to be a powerful tool for various

engineering design problems, and it has features which can be useful to deal with challenges that are difficult to handle using conventional network techniques.

## Materials and methods

### Process graph

The P-graph framework is a graph theoretic approach to PNS problems encountered during the design of industrial plants [18] as previously mentioned. In such problems, the challenge is to determine a suitable (and preferably optimal) design for the manufacture of a set of products, given available raw materials, capital, and component process units. The P-graph framework (e.g. Fig 1) uses a bipartite graph consisting of M-type nodes (circles) to represent materials, and O-type nodes (horizontal bars) to represent operating units; arcs are then used to signify how materials relate to processes as either inputs or outputs. For example, using Fig 1, O-type node 'O1' needs M-type node 'A' in order to provide M-type node 'E'. If the M-type node is produced by more than one O-type nodes, it can be produced by one, by another one, or any combination of them. For example, M-type node 'E' can be produced by O-type node 'O1', 'O2', or both of them. In a P-graph, inputs to an O-type node are necessary. For example, M-type nodes 'E' and 'F' are both necessary inputs to O-type node 'O4'. This makes O-type nodes O2 and O3 essential. As discussed below, these concepts can be adapted to represent ecological networks.

The P-graph framework consists of three component algorithms developed rigorously based on five axioms that serve as core assumptions of all PNS problems [18]. The axioms can be modified as shown in Table 1 to fit the context of ecological modelling. An ecosystem is

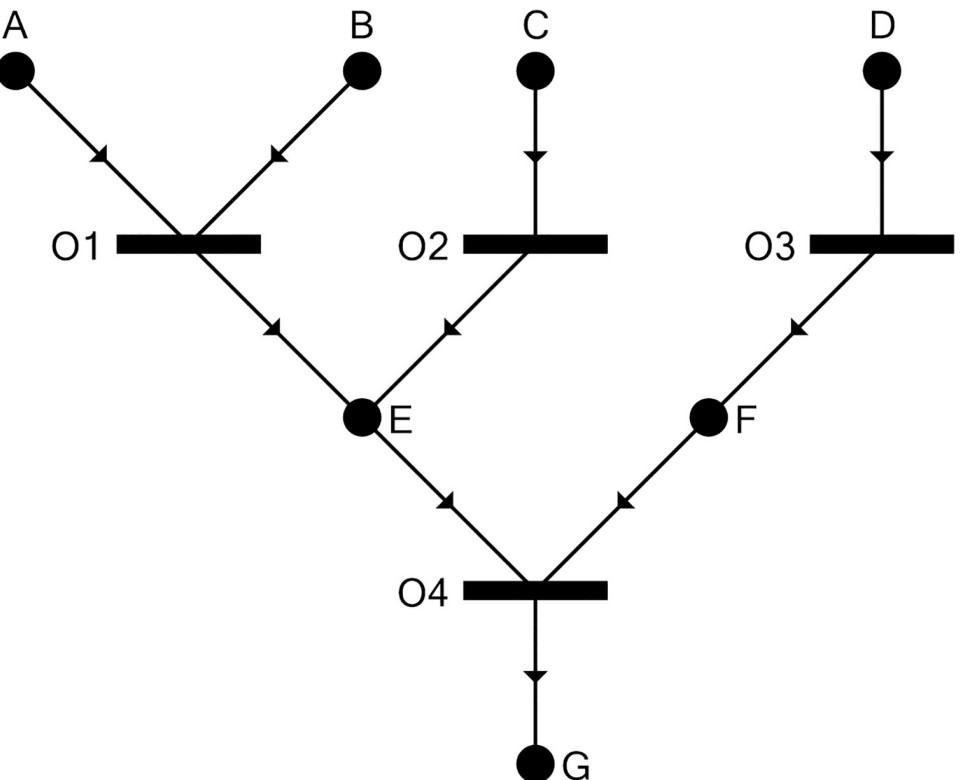

**Fig 1. P-graph representation of the process structure (lifted from [24]).**

**Table 1. P-graph axioms.**

| Process Engineering | Socio-Ecological Systems |
|---|---|
| (S1) Every final product is represented in the structure. | (SE1) There should be at least one well-defined terminal ecosystem service in the ecosystem structure. |
| (S2) A material represented in the structure is a raw material if and only if it is not an output of any operating unit represented in the structure. | (SE2) An ecosystem service represented in the structure is exogenous if and only if it is not an output of a functional unit defined in the ecosystem structure. |
| (S3) Every operating unit represented in the structure is defined in the synthesis problem. | (SE3) Every ecosystem functional unit in the ecosystem structure is well defined. |
| (S4) Any operating unit represented in the structure has at least one path leading to a product. | (SE4) Any ecosystem functional unit has at least one path leading to a terminal ecosystem service. |
| (S5) If a material belongs to the structure, it must be an input to or output from at least one operating unit represented in the structure. | (SE5) If an ecosystem service belongs to the ecosystem structure, it must be an input to or output from at least one ecosystem functional unit represented in the structure. |
| **Assumption**: capital is available to pay for operating units and operating costs. | **Assumption**: energy is available to keep the ecosystem structure functioning. |
| **Goals**: (1) meet production goal at (2) minimum cost for the structure and operation. | **Goals**: meet ecosystem services goal; and (2) minimize a cost metric (e.g. money, ecological footprint, energy, etc.) for the structure, management, functional, and operations of the ecosystem services. |

assumed to consist of components known as ecosystem functional units, which are analogous to operating units or processes in engineering design problems. These ecosystem functional units are represented as O-type nodes whose properties are based on current scientific knowledge on their local interactions. On the other hand, various ecosystem services that are analogous to material streams in engineering design problems are all represented as M-type nodes. Some ecosystem services can be classified as exogenous (i.e., originating entirely from outside the ecosystem boundary) and are analogous to raw materials for industrial plants; other ecosystem services are classified as terminal (i.e., exiting the ecosystem boundary to be utilized by humans) and correspond to industrial final products. Ecosystem services that are neither exogenous nor terminal act as intermediates and are produced and consumed entirely within the ecosystem network without crossing its boundaries.

The MSG algorithm rigorously generates a network which is the union of all structurally or combinatorially feasible networks that can be generated from the component units [25]. The SSG is capable of generating all combinatorially feasible networks, based on total enumeration of possible structures arising from localized relationships between materials and processes or services and functional units in the case of ecosystems [19]. Combinatorial feasibility is based solely on network connectivity and does not account for stream flow rates or quantities. The accelerated branch-and-bound algorithm (ABB) can then be used to effectively determine optimal and near-optimal solutions to PNS problems by evaluation of the performance of candidate networks, while excluding infeasible and redundant solutions from the search strategy. This effectiveness is derived from the use of implicit information embedded in all PNS problems [26]. A recent review paper gives a comprehensive survey of P-graph literature focusing on engineering applications [20]. The potential applicability of this framework to deal with PNS-like problems in generalized networks—such as economic systems or organizational structures—is discussed by [27]. In this study, the focus is purely on structural aspects and does not delve into stream flow rates.

P-graph can be used to perform a combinatorially complete search of ecological interactions to propose candidate ecological networks to represent a real ecosystem. The search procedure will automatically consider structural feasibility due to the ecological interactions. The

resulting structures can serve as skeletal frameworks for subsequent detailed analysis that considers the magnitudes of the flows and interactions. By imposing constraints, the search algorithm allows us to compose feasible pathways. These constraints include mass balances and the existence of known ecological interactions such as pollination or provision of shelter. The search domain to find feasible ecological interactions in the ecosystem can be reduced by eliminating or replacing species/entities. No feasible ecological interactions are lost in this process.

In practical ecosystem management, it is necessary to identify critical components (e.g., keystone species) to enable prioritization of limited resources. The P-graph methodology can be used as a basis to derive new metrics. Here, we propose a *criticality index* which measures the importance of a particular ecosystem component to the whole system, as quantified by the frequency of its occurrence in the alternative networks enumerated with SSG. The criticality index assumes a value in the interval [0, 1]. An index 0 signifies a component which is unnecessary or expendable—at least from the perspective of providing ecosystem services to an external human population. In other words, a component with minimum criticality may or may not be present, without affecting the viability of the entire network. At the opposite extreme are functional components with criticality indices of 1. They occur in all viable ecosystems and are thus indispensable components that provide essential ecosystems for which there are no alternative sources. The criticality index ($IC_i$) for functional component or species $i$ is given by,

$$IC_i = \frac{N_i}{N} \tag{1}$$

where $N_i$ is the number of alternate structures containing functional component $i$, and $N$ is the total number of alternate viable structures under consideration. This index provides a quantitative measure of the importance of each ecosystem component based solely on network properties, and does not account for any exogenous heuristic information. Alternative approaches can be used to account for subjective judgements that play a role in real-life ecosystem management. Techniques such as the analytic hierarchy process (AHP) can be used to systematically elicit weights from decision maker preferences [28].

## P-graph software and prototype

The P-graph software, called as *P-graph Studio*, is hosted and supported by the Department of Computer Science and Systems Technology at the University of Pannonia [24]. *P-graph Studio* can be accessed and used free of charge for research purposes via www.p-graph.org. It is currently on version 5.2.2.2 and can only run under Microsoft Windows. The following specifications are required: Microsoft.NET Framework 4.5.1 (x86 and x64) and Windows Installer 4.5. The website also contains a brief tutorial on P-graph fundamentals. Using the attached file in S1 Appendix, interested readers can replicate the case study that follows.

Since *P-graph Studio* is designed for use on engineering design problems, this article comes with a prototype code implemented in Microsoft Excel and Visual Basic for Windows (VBA) which is more readily used for the analysis of social-ecological networks. The prototype is made freely available here and can be retrieved from S2 Appendix. Users can replicate the case study that follows, modify it, or even analyze new examples.

## Results

### A simple socio-ecological system

This section illustrates the use of P-graph on socio-ecological systems via a case study of a stylized ecosystem interacting with a population of hunter-gatherers that rely on it as a food

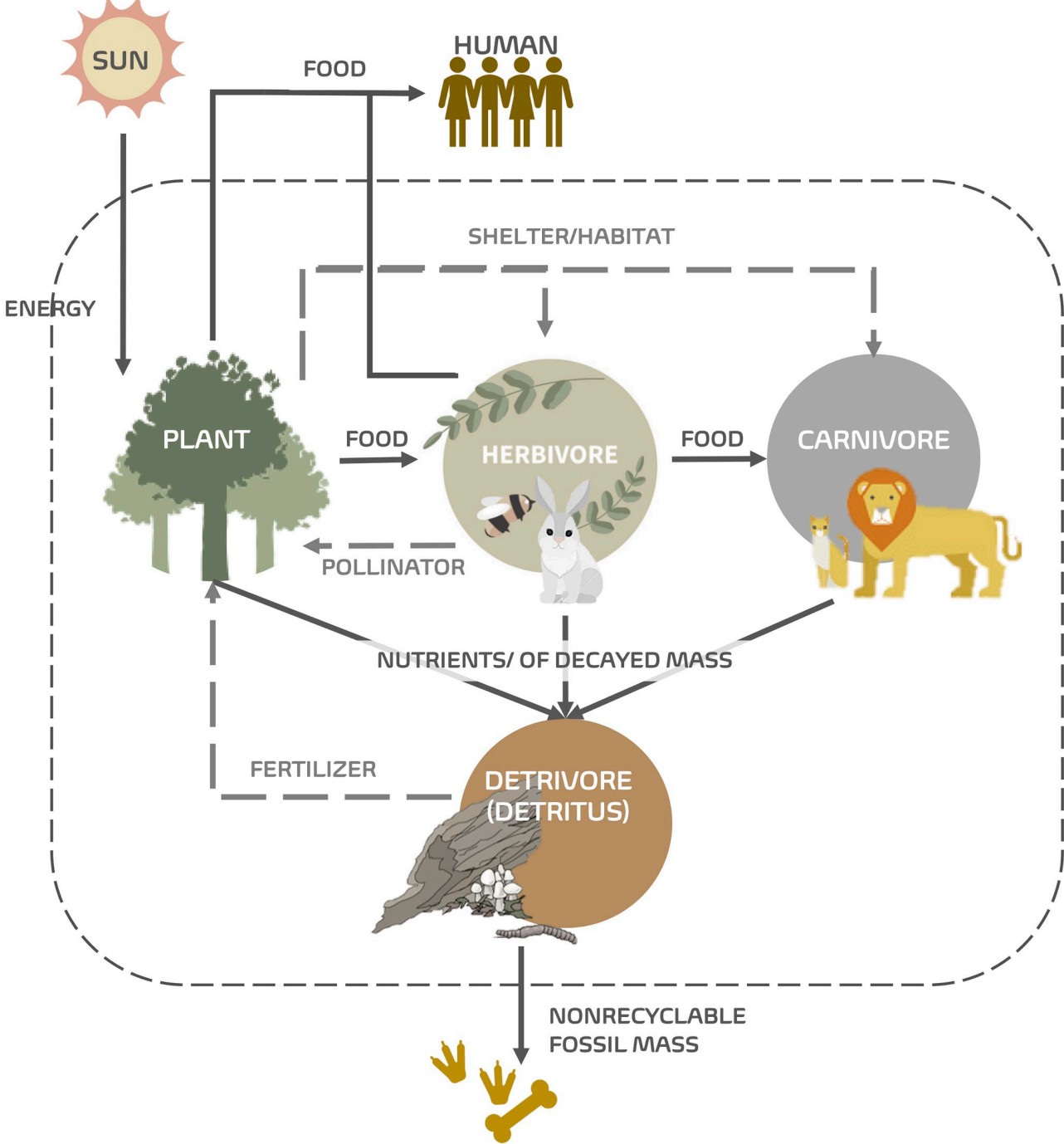

**Fig 2. The flow of energy through a simple ecosystem.** The energy flow from the sun is taken up by plants, some of which are eaten by herbivores, which, in turn, are eaten by carnivores. Part of the energy flows to the outside of the system.

source. The analysis shown here focuses purely on structural aspects and does not delve into stream flow rates. Energy usually enters ecosystems as sunlight and is captured in chemical form by photosynthesizes like plants and algae (see Fig 2). The energy is then passed through the ecosystem, changing form as organisms metabolize, produce waste, consume one another,

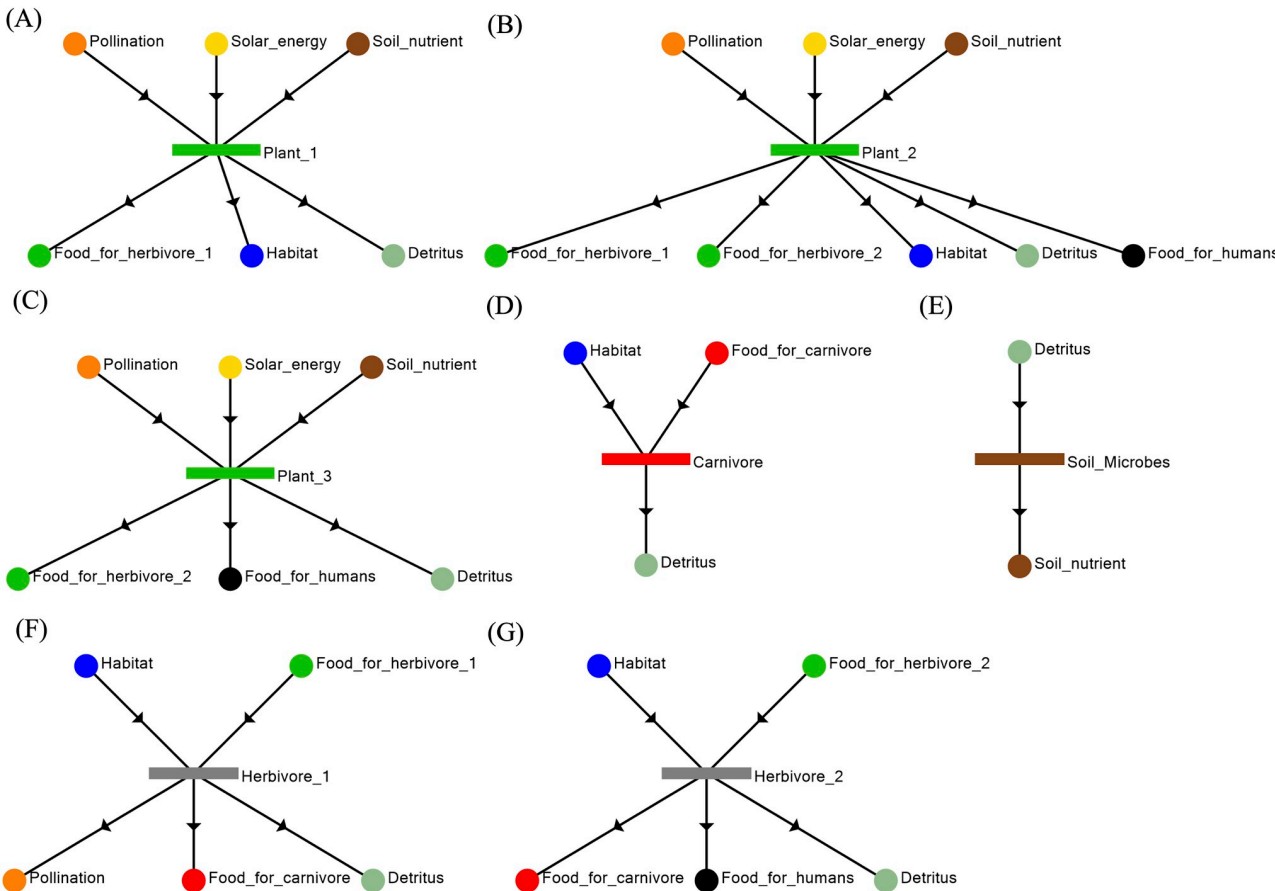

**Fig 3. Local interactions of the ecosystem components.** The ecosystem functional units presented here also corresponds to those listed in Table 2. It includes `Plant1` (A), `Plant2` (B), `Plant3` (C), `Carnivore` (D), `Soil_microbes` (E), `Herbivore1`(F), and `Herbivore2` (G).

and eventually, die and decompose. Each time energy changes forms, some of it is converted to heat as required by the second law of thermodynamics. Heat still counts as energy—and thus no energy has been destroyed—but it generally cannot be used as an energy source by living organisms. Ultimately, energy that enters the ecosystem as sunlight is dissipated as heat and radiated back into space on global scale. This one-way flow of energy through ecosystems means that every ecosystem needs a constant supply of energy, usually from the sun, in order to maintain its order and function. Energy can be passed between organisms, but it cannot be completely recycled because some of it is lost as heat in each transfer.

A socio-ecological system can be visualized as shown in Fig 2. In practice, the network representing such a system needs to be deduced from the local properties of its components or building blocks, as shown in Fig 3. In this system, we have defined specific characteristics for the plants and herbivores. For the plants, Plant 1 provides food for Herbivore 1, Habitat for herbivores and carnivores, and nutrients for Detritus (Fig 3A); Plant 2 not only provides food for Herbivore 1, Herbivore 2, and Humans, but also contributes to the Detritus and Habitat (Fig 3B); and Plant 3 contributes to the Detritus, and provides food for Herbivore 2 and Humans (Fig 3C). As for herbivores, both Herbivore 1 (Fig 3F) and Herbivore 2 (Fig 3G) contribute to the Detritus and provide food for Carnivore. Only Herbivore 1 contributes to the pollination, and only Herbivore 2 provides food for humans.

## Socio-ecological P-graph (network) structure

Based on the ecosystem functional units specified in Fig 3, MSG can deduce a maximal structure representing the ecosystem as shown in Fig 4A; the latter can be translated into box diagram shown in Fig 4B. The different ecosystem components are represented as separate O-type nodes. Whereas, the various ecosystem services are represented by M-type nodes. In practice, given the knowledge of local interactions of these components in Fig 3, the global structure of the ecosystem network may not be known a priori. However with MSG and SSG, they can be used to algorithmically assemble the components into a maximal structure and determine every structurally feasible network from the maximal structure for further analysis. The representation of final output of the ecosystem (i.e. food extracted by humans) illustrates the use of Axiom (SE1) in Table 1. Likewise, representation of solar energy as an external input is based on Axiom (SE2). All compartments (`Plant1`, `Plant2`, `Plant3`, `Herbivore1`, `Herbivore2`, `Carnivore`, `Soil_microbes`) are specified in the model as O-type nodes at the relevant level of resolution, based on Axiom (SE3). The inputs and outputs into each of these compartments signify their requirements and their contributions to the ecosystem, respectively, and they are represented as M-type nodes (`Habitat`, `Pollination`, `Soil_nutrients`, `Detritus`, `Food_for_Carnivore`, `Food_for_Herbivore1`, `Food_for_Herbivore2`). For example, herbivores draw on plant matter for food and also need shelter provided by vegetation in their habitat; in turn, they provide food to carnivores and humans, and also propagate seeds of plants. These inputs and outputs into the compartments are defined based only on local information; generation of the ecosystem network can be done algorithmically in P-graph via MSG. It can easily be seen in Fig 4A that the given structure also satisfies Axioms (SE4) and (SE5) in Table 1. That means that each functional unit has a path or contribution to the terminal ecosystem services (SE4), and all ecosystem services are inputs or outputs from a functional unit.

## Discussion

In this case study, the true structure of the ecosystem is given by the maximal structure (Fig 4A). SSG can thus be used to generate all nominally functional ecosystems (i.e., those which are structurally able to support the population of hunter-gatherers). Enumeration of these alternative networks can then tell us which ecosystem components are valuable from the perspective of the human population. Given the P-graph structure in Fig 4A, twenty structures (S1-S20) can be generated. They are summarized in Table 2 and are illustrated in supplementary information S1 Fig. Each ecosystem functional unit in Table 2 is visualized by the graphs in Fig 3.

Interestingly, it can be observed that for this particular ecosystem, `Carnivore` is not essential in the sense that the rest of the ecosystem is capable of producing the terminal services without it. This can be observed if the structures in Table 2 are compared pairwise—the odd-numbered structures contain `Carnivore` whereas the even-numbered structures do not. In other words, with or without `Carnivore`, the ecosystem seems to still function. Unlike the plants (`Plant1`, `Plant2`, `Plant3`) and herbivores (`Herbivore1`, `Herbivore2`), there is only one type of carnivore defined in the system. Without `Carnivore`, the node representing `Food_for_carnivores` will be reclassified as a terminal component, which may lead to overpopulation of some herbivores. The fact that `Carnivore` can be not essential is a consequence of excluding the carnivores as a source of `Food_for_humans`, hence removing their direct contribution to the terminal ecosystem service of `Food_for_humans`. Note that there is no arc connecting `Carnivore` to the `Food_for_humans`. Based on the defined

(A)

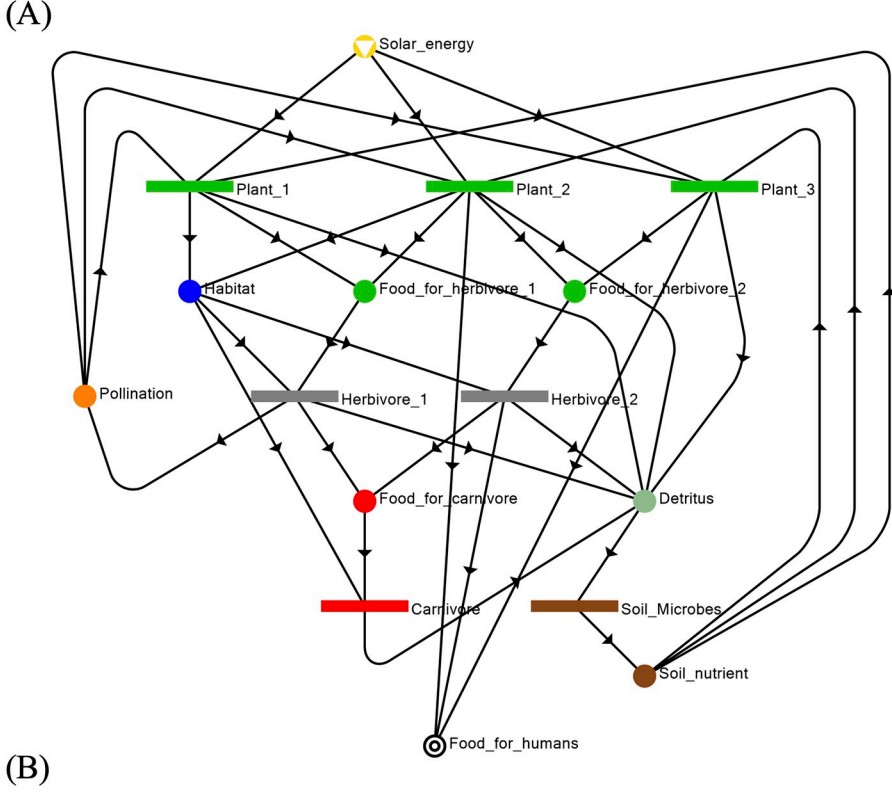

(B)

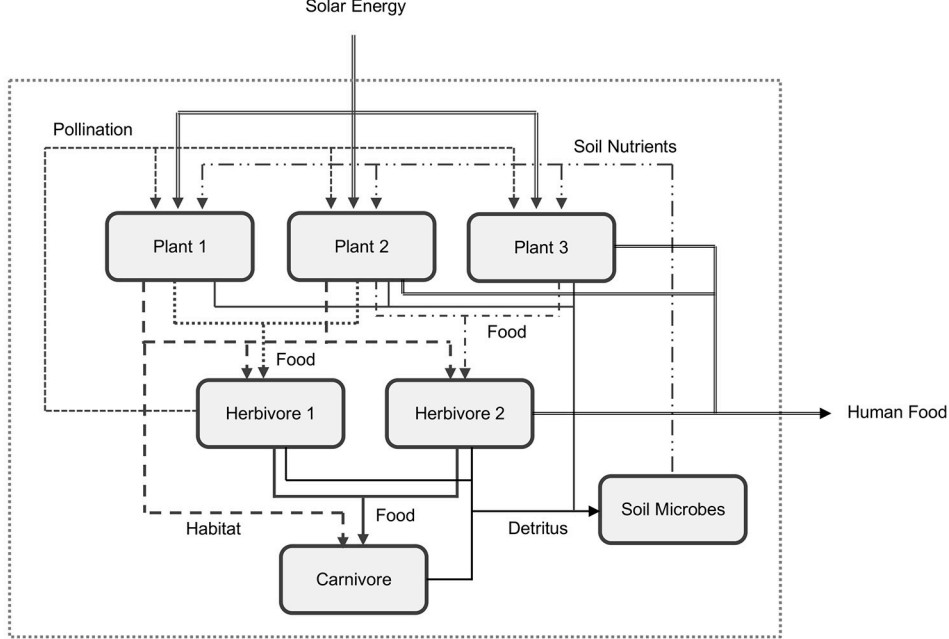

**Fig 4. P-graph of the socio-ecological system.** (A) Socio-ecological P-graph representing the union of the ecosystem functional units in Fig 3. This represents Structure S1, the maximal and most redundant structure in Table 2. (B) Box diagram of the extended socio-ecological system illustrated in (A).

**Table 2. The ecosystem functional units contained in each of the 20 structures.**

| Structure Label | Plant1 Fig 3A | Plant2 Fig 3B | Plant3 Fig 3C | Herbivore1 Fig 3F | Herbivore2 Fig 3G | Carnivore Fig 3D | Soil microbes Fig 3E |
|---|---|---|---|---|---|---|---|
| S1 | ✓ | ✓ | ✓ | ✓ | ✓ | ✓ | ✓ |
| S2 | ✓ | ✓ | ✓ | ✓ | ✓ |  | ✓ |
| S3 | ✓ | ✓ | ✓ | ✓ |  | ✓ | ✓ |
| S4 | ✓ | ✓ | ✓ | ✓ |  |  | ✓ |
| S5 | ✓ |  | ✓ | ✓ |  | ✓ | ✓ |
| S6 | ✓ |  | ✓ | ✓ |  |  | ✓ |
| S7 | ✓ |  | ✓ | ✓ | ✓ | ✓ | ✓ |
| S8 | ✓ |  | ✓ | ✓ | ✓ |  | ✓ |
| S9 | ✓ | ✓ |  | ✓ | ✓ | ✓ | ✓ |
| S10 | ✓ | ✓ |  | ✓ | ✓ |  | ✓ |
| S11 | ✓ | ✓ |  | ✓ |  | ✓ | ✓ |
| S12 | ✓ | ✓ |  | ✓ |  |  | ✓ |
| S13 |  | ✓ | ✓ | ✓ | ✓ | ✓ | ✓ |
| S14 |  | ✓ | ✓ | ✓ | ✓ |  | ✓ |
| S15 |  | ✓ | ✓ | ✓ |  | ✓ | ✓ |
| S16 |  | ✓ | ✓ | ✓ |  |  | ✓ |
| S17 |  | ✓ |  | ✓ | ✓ | ✓ | ✓ |
| S18 |  | ✓ |  | ✓ | ✓ |  | ✓ |
| S19 |  | ✓ |  | ✓ |  | ✓ | ✓ |
| S20 |  | ✓ |  | ✓ |  |  | ✓ |
| $N_i$ | 12 | 16 | 12 | 20 | 10 | 10 | 20 |
| $IC_i$ | 0.6 | 0.8 | 0.6 | 1 | 0.5 | 0.5 | 1 |

social-ecological system (Fig 2), if humans only eat plants and herbivores, the existence of the carnivores is not essential.

The maximal and minimal structures are S1 (Fig 4A is also Fig 1A of S1 Fig) and S20 (Fig 5B), respectively. The maximal structure S1 contains all the components and entities defined in the social-ecological system (Fig 4A). In the minimal structure S20 (shown in Fig 5B), `Plant1`, `Plant3`, `Herbivore2`, and `Carnivore` are excluded from the system. This is a "bare bones" system with the absolute minimum functional units necessary to produce the terminal service of `Food_for_humans`. Fig 5A shows structure S10 from Table 2 which is an intermediate between the maximal structure (S1) and the minimal structure (S20). It can be seeing that structure S10 has some redundancy to provide food to humans from plant P1 or herbivore H2. This will, therefore, give it some ability to manage perturbations such as for example, a decrease in the population of plant P2 which in the minimal structure(S20) is the only source of food for humans.

Finally, note that for the plants, at least one of `Plant1`, `Plant2`, and `Plant3` should be included in the system. If two plant types are to be excluded in the system at the same time, `Plant2` cannot be one of them. It can only be `Plant1` and `Plant3`, as shown in Table 2. Removing `Plant2` and `Plant3` will directly affects the food provided for humans. Whereas, removing `Plant1` and `Plant2` directly affect the food provided for Herbivore1 and the habitat provided for the herbivore and carnivore. All these will disrupt the sustainability of the system and management of the ecosystem services.

In general, a functional ecosystem structure needs to always contain the following: at least one plant type (`Plant1`, `Plant2`, `Plant3`), at least one herbivore type (`Herbivore1`,

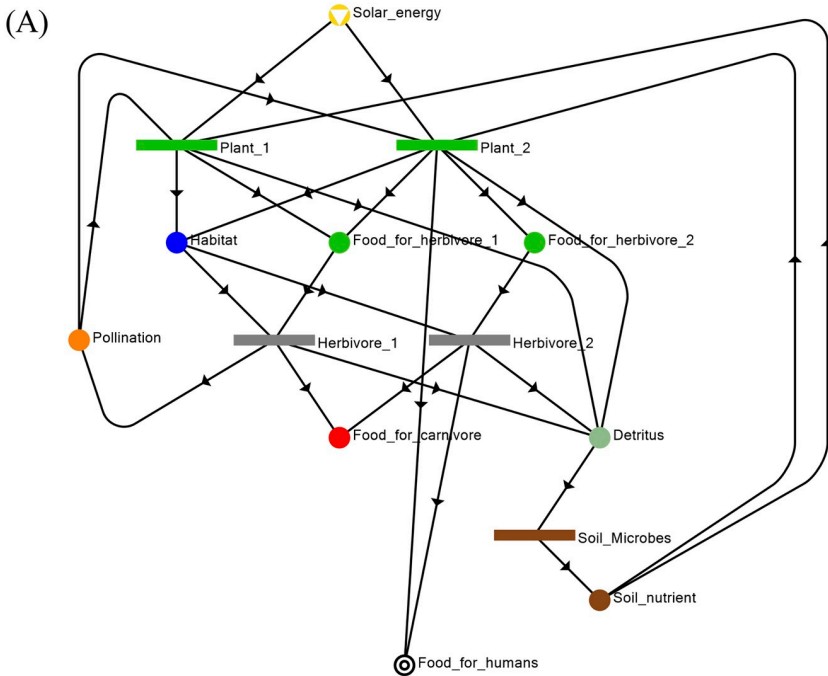

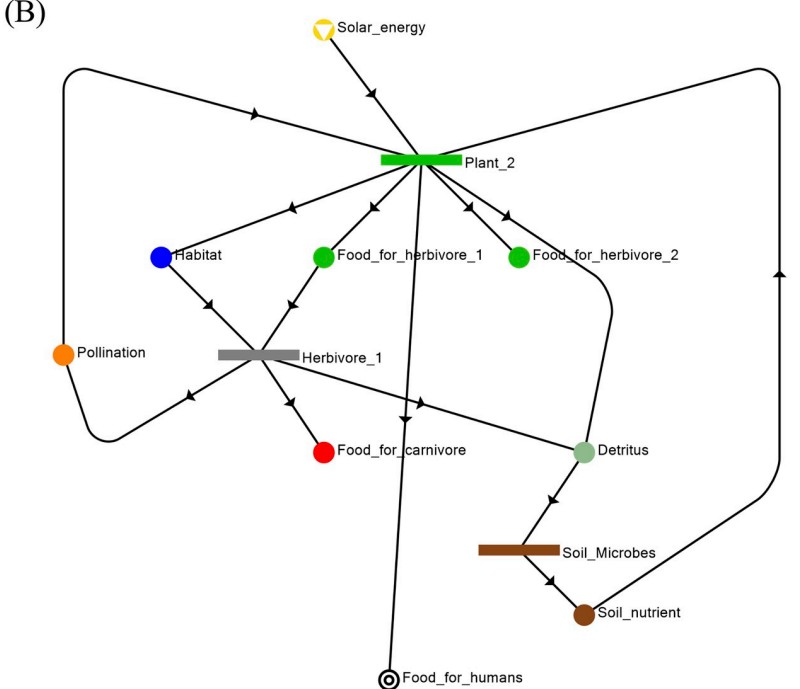

**Fig 5. Minimal structure generated from the P-graph illustrated in Fig 4.** The 20 structures generated from P-graph can be found in S1 Fig. (A) Structure S10 is an intermediate structure between the maximal structure (S1) and the minimal structure (S20). (B) The minimal structure is Structure S20 in Table 2, it does not need `Plant1`, `Plant2`, `Herbivore2`, and `Carnivore` to provide the necessary service.

`Herbivore2`), and the `soil_microbes`. However, both `Herbivore1` and `soil_microbes` are irreplaceable (they are present in all 20 structures) and appear to be crucial in the social-ecological system. It can be explained by the important components they provided, e.g. `Soil_nutrients` and `Pollination`, that are essential and needed by the plants (`Plant1`, `Plant2`, `Plant3`) to grow and live.

There is one key distinction between this new application of P-graph and its original purpose. In engineering PNS problems, the task is to generate possible networks from process building blocks as part of the design procedure. In this work, the ecosystem network already exists but the full structure may not be completely known to the human observer. P-graph thus serves two purposes. First, the overall network structure is deduced based on an understanding of the local properties of its components using MSG. Second, possible networks that can emerge from the present one are also deduced using SSG. These alternative networks represent the structurally viable ecosystems that can evolve from the current one due to human-induced or natural perturbations, e.g. the removal of a functional unit or biological species.

The capability to enumerate all combinatorially feasible networks or viable ecosystems via SSG also allows quantitative analysis of ecosystem components. Table 2 lists the $N = 20$ viable ecosystems for the illustrative case study. In Table 2, `Herbivore1` and `Soil_microbes` both have indices of 1, being unique providers of pollination services and soil nutrients, respectively. The other components in this ecosystem are of intermediate criticality. It can also be seen that the criticality of the individual components is dependent on the structural redundancy or degeneracy of the ecosystem network; a small number of critical components implies that the ecosystem is robust to disruptions. This means that system has the capacity to adjust to an alternate structure to compensate for the loss of a functional unit. Conversely, a large number of critical components implies that almost any loss of a component can be fatal to the overall system, i.e the system does not have any or has very few alternate structures to shift to compensate for the loss of functional units. This concept can be explored further in future developments. At this stage, the proposed criticality index can be used for the practical problem of prioritizing components for ecosystem management purposes. The question of the availability of feasible or viable structures that the system can shift to, gives rise to the concept of ecosystem brittleness or the converse concept of resilience. But this is an extensive topic which will be the subject of future research.

## Implications and prospects

The P-graph framework offers the prospect of extending the toolbox of ecological modelling, and brings two unique capabilities to bear. First, as a bipartite graph, P-graph allows unambiguous representation of multiple types of relationships that exist among compartments in a real ecosystem. This allows richer representation of ecosystems by enabling multiple roles and functions to be represented within a graph.

Secondly, P-graph allows error-free generation of a complete network model based on local information about system components (via MSG), and furthermore allows all structurally feasible subsets of the complete network to be enumerated (via SSG). Such a capability overcomes the need to rely on heuristic assembly rules that are currently prevalent in ecological network analysis. The capacity for complete enumeration of functional ecosystems also allows a new criticality index to be determined for each ecosystem component, thus adding a new concept to the established ecological network metrics currently in use [29].

The simple case study shown here focuses purely on structural features based on the existence of multi-functional links among compartments. As the results of the example illustrate, these structural features can be used to identify vital but indirect linkages among ecosystem

components, which would otherwise be difficult to detect by direct inspection. Unlike most ecological network analysis techniques that identify keystone species based on trophic relationships [30], P-graph can establish the importance of ecosystem components based on multiple roles that they play. The approach can also be extended further to integrate ecological stoichiometry concepts [31], provided that the appropriate ratios are known. A P-graph model can also serve as a structural framework to subsequently develop a dynamic model of an ecosystem.

In addition to its stand-alone capabilities, the P-graph framework can also be used in conjunction with other concepts and tools that are relevant to the effective management of ecosystems. Biodiversity loss and ecosystem collapse are major environmental issues; understanding emergent properties of ecological networks can provide valuable insights for retarding or stopping irreversible damage. The concept of resilience has become an important aspect in the design of different man-made networks which need to be designed to withstand shocks [32]. Resilient systems need to be planned to be able to absorb, recover from, and adapt to perturbations at the network level [33]. The concept can be extended to managing ecosystems to allow them to withstand pressure from climate change, human encroachment, and disruptive events such as natural disasters. In PNS problems, P-graph methodology has been shown to be effective for designing robust and resilient systems via process-level redundancy and network level degeneracy [34]. This capability can also be readily extended to PNS-like problems in ecological network analysis, so that rational and effective plans can be developed for the protection of ecosystems.

The relationship between P-graph representations of ecosystems and the aforementioned multiplex networks also needs further examination. In [23], Hutchinson et al. identify as an important research challenge the process of deducing the structure of ecological multilayer networks from characteristics of species. It has been shown that P-graph can deduce all combinatorially feasible ecological network structures that can be formed from a set of building blocks (i.e., species or compartments). Thus, if the equivalence between P-graphs and multiplex graphs can be rigorously established in the future, then the MSG and SSG algorithms can also be used to generate multilayer networks based on knowledge of the local interactions of each species. This possibility is intriguing, but remains speculative at this stage. Further research is needed to establish if the two mapping approaches are indeed equivalent.

## Conclusion

We propose the extension of the P-graph framework to ecosystem network structure and modeling. This novel application brings to bear specific features of P-graph to overcome the limitations of currently available tools (e.g., ecological network analysis). In particular, this framework allows (a) simultaneous representation of multiple types of relationships among ecosystem compartments; (b) rigorous generation of complete network models via MSG; and (c) elucidation of different feasible or functional ecosystems structures via SSG. The P-graph thus enables new insights to be drawn for purposes of effective management of ecosystem services, as illustrated here via a stylized ecosystem case study.

Further opportunities can be explored by using the optimization capability of P-graph for specific types of ecological modelling problems. The bulk of ecological network analysis literature focuses on the use of descriptive measures to understand the relative importance of compartments. However, in some problems, a specific objective function can be identified; for example, in the context of managing an ecosystem which is being altered or is being altered. This is important because ecosystems across the globe are being altered by human activity. The work may also be of scientific or policy interest to determine the maximum sustainable rate of

withdrawal of renewable resources or ecosystem services from the environment. This case is a common issue in fisheries management. Conversely, there are cases where it will be necessary to determine the minimum land area needed to sustain a functional ecosystem, in response to loss of area due to habitat destruction by humans or by sea level rise. Other optimization problems in ecological modelling can be identified and solved in the future with the aid of P-graph via the MSG algorithm.

There are interesting prospects for future research in this new sub-area of systems ecology. The most promising directions are the direct application of the methodology to real ecosystems, and the hybridization of P-graph with established techniques such as ecological network analysis, Petri net and multilayer network to enable better understanding of the behavior of ecological networks.

## Supporting information

**S1 Appendix. P-studio file.** P-graph generated from P-studio.
(PGSX)

**S2 Appendix. P-graph Excel/VBA macros.** P-graph VBA macro file to generate MSG & SSG.
(XLSM)

**S3 Appendix. Brief introduction of MSG and SSG algorithms.** It contains the pseudocode and brief description of the MSG and SSG algorithms.
(PDF)

**S1 Fig. Structures generated from the P-graph.** Twenty P-graph structures including MSG & SSG.
(PDF)

## Acknowledgments

The authors would like to acknowledge the Pázmány Péter Catholic University for proving the necessary facilities which allowed for the conduct of discussions among the authors of this manuscript.

## Author Contributions

**Conceptualization:** Angelyn Lao, Heriberto Cabezas, Raymond Tan.

**Data curation:** Ákos Orosz.

**Formal analysis:** Angelyn Lao, Raymond Tan.

**Investigation:** Angelyn Lao, Heriberto Cabezas, Raymond Tan.

**Methodology:** Angelyn Lao, Heriberto Cabezas, Raymond Tan.

**Resources:** Ákos Orosz.

**Supervision:** Angelyn Lao, Heriberto Cabezas, Raymond Tan.

**Validation:** Angelyn Lao, Heriberto Cabezas, Ferenc Friedler, Raymond Tan.

**Visualization:** Angelyn Lao, Heriberto Cabezas, Ákos Orosz.

**Writing – original draft:** Angelyn Lao, Heriberto Cabezas, Raymond Tan.

**Writing – review & editing:** Angelyn Lao, Heriberto Cabezas, Ferenc Friedler, Raymond Tan.

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
