## [Decision Letter · Decision Letter 0]

18 May 2020

PONE-D-20-10193

Socio-Ecological Network Structures from  Process Graphs

PLOS ONE

Dear Dr. Lao,

Thank you for submitting your manuscript to PLOS ONE. After careful consideration, we feel that it has merit but does not fully meet PLOS ONE’s publication criteria as it currently stands. Therefore, we invite you to submit a revised version of the manuscript that addresses the points raised during the review process.

Reviewers clearly disagree in their assessment.  Please address the comments to the extent possible.  Also, interconnectedness of socio-ecological systems is discussed in the field of resilience, especially in papers that view system resilience as property of interconnected networks (e.g. Ganin et al., (2016). Operational resilience: concepts, design and analysis. *Nature Scientific Reports, ***6(1); L**inkov, I., **& Trump, B. D. (2019). *The Science and Practice of Resilience. *Springer, Amsterdam.). **Discussion of these issues may be helpful for positioning the paper

We would appreciate receiving your revised manuscript by Jul 02 2020 11:59PM. To enhance the reproducibility of your results, we recommend that if applicable you deposit your laboratory protocols in protocols.io, where a protocol can be assigned its own identifier (DOI) such that it can be cited independently in the future. For instructions see: http://journals.plos.org/plosone/s/submission-guidelines#loc-laboratory-protocols

We look forward to receiving your revised manuscript.

Kind regards,

Igor Linkov

Academic Editor

PLOS ONE

Journal Requirements:

Reviewers' comments:

Reviewer's Responses to Questions

**Comments to the Author**

1. Is the manuscript technically sound, and do the data support the conclusions?

Reviewer #1: Yes

Reviewer #2: Partly

2. Has the statistical analysis been performed appropriately and rigorously? 

Reviewer #1: N/A

Reviewer #2: N/A

3. Have the authors made all data underlying the findings in their manuscript fully available?

Reviewer #1: Yes

Reviewer #2: No

4. Is the manuscript presented in an intelligible fashion and written in standard English?

Reviewer #1: Yes

Reviewer #2: Yes

5. Review Comments to the Author

Reviewer #1: This is a paper suggesting an alternative approach to ecological network analysis (ENA), namely the process graph (or P-graph) one. The authors draw this methodology from the industrial applications’ field and argue that it is superiour to traditional ENA as it separates every operational and intermediate component of such a network explicitly and gives insights of its viable realizations out of all its possible ones. More importantly, it has a say on the critical components of such systems, thus elucidating managing aspects of those systems. Lastly, the framework is neatly presented in a generic example rising from first principle, ecological arguments. The example is reproducible with code provided in Visual Basic and Excel macros.

The work is coherent and consistent with the established methods of P-graphs and has value to at least the audience of ecology. I am therefore confident that it is suitable for publication in PLOS One and will be an asset to the community.

As some thoughts and secondary remarks however, I would advise the authors to consider the following points prior to finalising the publication of this paper:

1) Petri nets seem to have a lot of common properties with P-graphs. What is the connection of ecological P-graphs to Petri nets? How are the two different? Broaching this topic in the text would broaden the work’s scope.

2) It is implicit but poorly described in the materials and methods section that the Accelerated-Branch-and-Bound (ABB) algorithm comes into the analysis flow and process information, which, in a 3-level procedure, makes the application of the scheme MSG-→SSG-→ABB for ecological networks sensible.

3) Since this is a recurrent theme throughout the paper, an example or a procedural (or algorithmic) explanation of the “local interactions” giving rise to the MSG would be useful for the reader who is not familiar with the algorithm. Especially illustrative would be to draw a connection of the local interactions with an ecosystem’s example.

4) As specified in the text, plant 2 is necessary to ensure the “food for the humans” terminal. This is owed to the weave of interconnections following the intermediate operational units and nodes as resulting from the original assignment of the system’s components to O and M nodes and the P-graph’s algorithms. I find this to be a remarkable result with lots of extensions for practical ecosystem applications and would advise it to be further elaborated and highlighted.

5) The way the numerator of the criticality index is explained might cause some confusion as to whether the systems referred to are viable or not. I would add this explicitly.

6) Although nothing significant through a superficial read, there are some minimal language slips and therefore a proof-reading would eliminate the few issues standing out.

Reviewer #2: # Summary

The authors' manuscript applies the process graph (P-graph) approach

from industrial plant design to ecological systems. The ecological

application of the P-graph is proposed as a bipartite representation

of an ecosystem in which processes and ecosystem compartments are two

distinct parts in the network. All compartments are "indirectly"

connected via processes. The authors' state that the P-graph approach

has several advantages over current ecological network modeling

approaches. The main being that the use of the bipartite

(process-compartment) structure, permits compartments to have multiple

roles in the ecosystem, such as both prey and pollinator. The majority

of ENA models only consist of a single "currency", typically biomass

or nutrients. The authors also present how the identification of

"maximal" and "minimal" network structures, relative to a desired

output from the ecosystem, permits the quantification of the effects

of species loss, efficacy of reconstruction efforts and maximum

sustainable exploitation by humans via the calculation of a

criticality index. The authors present an application of the method to

a "stylized" example network and conclude that the P-graph method is a

novel approach that can provide a useful tool for solving ecological

optimization problems in the context of increasing global pressures on

the environment.

# Comments

My primary concern is that the P-graph method appears to be a special

case of a multi-layer graph approach, which has already been published

widely in ecology. As such, I do not see the claims of novelty

justified without a more complete graph theoretic analysis of the

P-graph method as it compares to multi-layer networks. In particular,

two high profile articles have already been published that provide

developments in this area that are not cited:

Pilosof, S., Porter, M., Pascual, M. et al. The multilayer nature of

ecological networks. Nat Ecol Evol 1, 0101 (2017).

Matthew C. Hutchinson Bernat Bramon Mora Shai Pilosof Allison

K. Barner Sonia Kéfi Elisa Thébault Pedro Jordano Daniel B. Stouffer

Seeing the forest for the trees: Putting multilayer networks to work

for community ecology. Func. Ecol. 33, 2 (2018).

One specific criticism is that the P-graph approach and associated

metrics, as currently described, assume an un-weighted graph, which is

a major step back from the multi-layer graph methods that already

incorporate weighted and signed edge values. In addition, multilayer

analyses have been implemented in python and are available across all

major operating systems that run Python, as opposed to p-graph, which

is only available on Windows.

Within the context of P-graph and how it is presented, the MSG and SSG

algorithms are not described and I didn't see references given that

describe them. As such I have indicated above that not all data have

been provided, as this is a core component of the paper. Where they

are defined in the text, the citation is for a general ENA modeling

method by Fath et al. that doesn't discuss these algorithms. This

should be corrected by either adding such a reference or including a

description of the algorithms here.

The MSG method assumes the described P-graph as the maximal network;

however, many ecosystems of interest are in a state of human induced

disturbance and may represent a reduced state of the ecosystem, as is

acknowledged by the authors. How can such assumptions about the

maximal network be accounted for when applying such analyses to the

management of real ecosystems?

Every ecosystem functional unit is well defined is an axiom of the

p-graph method. This would be possible in a controlled production

setting of a factory but is unrealistic in the vast majority of

ecosystems both due to the numerous functions and because of

behavioral and evolutionary variability. Given these as potential

issues, what is the consequence for the P-graph approach?

There is a discussion point that states a product of the analysis

being that given no direct link from carnivores to human food,

carnivores would be determined to be non-essential. This seems to

point to an issue with this approach and how it is not accounting for

indirect effects. For example, what is the indirect effect of

carnivores on human food production via the nutrient inputs from

carnivores both directly and indirectly through predation? Perhaps

this is an issue with interpretation of the analysis, but, if this

were used as a management tool, under this interpretation, carnivores

might be incorrectly deemed lower in importance relative to human food

production than they are in actuality.

With regard to the criticality index, there is a significant

limitation resulting from the lack of a weighted metric in that

sufficient quantities required for functions are not quantified. The

un-weighted graph assumes equivalence and implies sufficiency to meet

the requirements of each function. Continuing with the carnivore

example, in a real ecosystem as a sequential extirpation of prey

species were to proceed the availability of prey would eventually

reach a point at which there would be an insufficient amount to

sustain a viable population of carnivores. That level might be reached

prior to having lost all herbivore species, especially if the

abundances of herbivores is un-even, as is often the case in

ecosystems. This and similar issues require greater clarification

prior to recommending such a tool as a useful resource for ecosystem

management decisions.

Given that society is only a final consumer in the example models, and

is strongly implied in the list of P-graph axioms, I struggle to see

how the term socio-ecological applies to this approach as represented

here. Does the "socio-" in socio-ecological refer to human society

here? If so, does the P-graph approach have greater potential for

integrating more dimensions of human interactions with ecosystems? If

not, is "socio-" referring to the "sociological" nature of non-trophic

interactions (e.g. pollination)?

Per the statement about the solution of optimization problems on Line

340, the use of the term optimization implies maximizing a goal

function with regard to costs. As such, this terminology inherently

supports a perspective that overlooks the need for capacity and

measures to facilitate resilience in the face of disturbance. Although

the p-graph approach incorporates increased complexity of systems

relative to many other network approaches by accommodating the analysis

of different functions in a single network, it does not incorporate

important aspects of ecosystems that aid in resilience, such as

adaptation and evolutionary dynamics. As I already mentioned above

with regard to the carnivore example, variables such as minimal viable

population size, are essential to the prevention of species

loss. Can these dimensions of ecosystem dynamics somehow be accounted

for with the P-graph approach? If not, how can the results of the

P-graph analyses be qualified in light of such information?

6. PLOS authors have the option to publish the peer review history of their article (what does this mean?). If published, this will include your full peer review and any attached files.

Reviewer #1: Yes: Dr. Jason Bassett

Reviewer #2: Yes: Matthew Kekoa Lau

---

## [Author Response · Author response to Decision Letter 0]

19 Jun 2020

Dear Editor-in-Chief Dr. Heber,

We have uploaded the revised version of our manuscript entitled “Socio-Ecological Network Structures from Process Graphs”, and our rebuttal to Reviewer 1 and Reviewer 2.

Our general assessment is that most of the criticisms (especially from Reviewer 2) arise from a lack of a clear understanding of the capabilities and limitations of the P-graph framework. Thus, we have revised the manuscript to give a clearer picture of the class of problems that our work addresses via P-graph. We also discuss the possible interface of P-graph with other more well-known tools.

We strongly believe that we have successfully addressed all concerns raised by the reviewers and we hope that you will accept this manuscript for publication in PLOS One.

Sincerely,

Angelyn Lao, Heriberto Cabezas, Ferenc Friedler, Ákos Orosz, and Raymond R. Tan

---

## [Decision Letter · Decision Letter 1]

16 Jul 2020

PONE-D-20-10193R1

Socio-Ecological Network Structures from  Process Graphs

PLOS ONE

Dear Dr. Lao,

Thank you for submitting your manuscript to PLOS ONE. After careful consideration, we feel that it has merit but does not fully meet PLOS ONE’s publication criteria as it currently stands. Therefore, we invite you to submit a revised version of the manuscript that addresses the points raised during the review process.

We look forward to receiving your revised manuscript.

Kind regards,

Igor Linkov

Academic Editor

PLOS ONE

Reviewers' comments:

Reviewer's Responses to Questions

**Comments to the Author**

1. If the authors have adequately addressed your comments raised in a previous round of review and you feel that this manuscript is now acceptable for publication, you may indicate that here to bypass the “Comments to the Author” section, enter your conflict of interest statement in the “Confidential to Editor” section, and submit your "Accept" recommendation.

Reviewer #1: All comments have been addressed

Reviewer #2: All comments have been addressed

2. Is the manuscript technically sound, and do the data support the conclusions?

Reviewer #1: Yes

Reviewer #2: Yes

3. Has the statistical analysis been performed appropriately and rigorously? 

Reviewer #1: N/A

Reviewer #2: N/A

4. Have the authors made all data underlying the findings in their manuscript fully available?

Reviewer #1: Yes

Reviewer #2: Yes

5. Is the manuscript presented in an intelligible fashion and written in standard English?

Reviewer #1: Yes

Reviewer #2: Yes

6. Review Comments to the Author

Reviewer #1: I thank the authors for addressing the comments raised and wish them a smooth remainder of the publication process.

Reviewer #2: The authors have adequately resolved the main comments that I raised in the previous review.

In particular, making the algorithms for the main analyses available and tempering the claims of novelty with regard to multi-level network analysis and also Petri Nets (as requested by the other reviewer).

I would add two suggestions that would help to improve the clarity of the manuscript.

1. The development and analysis of the Criticality Index is done in the Discussion. I suggest moving this to the Methods and Results sections.

2. On this read, I realized that there is some confusion in terminology. Consider clarifying:

- "Conventional ENA", which seems to refer to Ecosystem Network Analysis that developed from input-output analysis, should be made explicit as the term ecological network analysis refers to a broad array of both modeling and analyses.

- P-graph the modeling and analysis technique and P-graph the software, sometimes referred to as P-graph Studio. I suggest making P-graph the software italicized or underlined, as titles often are.

7. PLOS authors have the option to publish the peer review history of their article (what does this mean?). If published, this will include your full peer review and any attached files.

Reviewer #1: No

Reviewer #2: **Yes: **Matthew Kekoa Lau

---

## [Author Response · Author response to Decision Letter 1]

18 Jul 2020

We have uploaded the point-by-point response document to the reviewers.

---

## [Editor Report · Decision Letter 2]

22 Jul 2020

Socio-Ecological Network Structures from  Process Graphs

PONE-D-20-10193R2

Dear Dr. Lao,

We’re pleased to inform you that your manuscript has been judged scientifically suitable for publication and will be formally accepted for publication once it meets all outstanding technical requirements.

Kind regards,

Igor Linkov

Academic Editor

PLOS ONE
---

## [Editor Report · Acceptance letter]

24 Jul 2020

PONE-D-20-10193R2 

Socio-Ecological Network Structures from Process Graphs 

Dear Dr. Lao:

I'm pleased to inform you that your manuscript has been deemed suitable for publication in PLOS ONE. Congratulations! Your manuscript is now with our production department. 

Kind regards, 

on behalf of

Dr Igor Linkov 

Academic Editor

PLOS ONE